# Mental Health Support in Higher Education during the COVID-19 Pandemic: A Case Study and Recommendations for Practice

**DOI:** 10.3390/ijerph20064969

**Published:** 2023-03-11

**Authors:** Alicja Lisiecka, Dorota Chimicz, Agnieszka Lewicka-Zelent

**Affiliations:** Faculty of Education and Psychology, Maria Curie-Skłodowska University, 20-400 Lublin, Poland

**Keywords:** COVID-19 pandemic, mental health, support, higher education

## Abstract

**Background:** The COVID-19 pandemic has caused changes in the lives of many university students around the globe, including students at Maria Curie-Skłodowska University in Lublin, Poland. Isolation, a sense of threat, and the transition to remote learning resulted in numerous, mainly psychological, negative consequences for students. The university aimed to provide students with effective assistance during the COVID-19 pandemic. The question now is whether it has succeeded or failed. This study demonstrates good practices in mental health support at Maria Curie-Skłodowska University during the pandemic and post-pandemic period. **Methods:** The study was conducted between October and December 2022. The case study method and purposive sampling were used in the study. A total of 19 participants took part in this study. Of the respondents, 16 were females, 3 were males. Ages ranged between 26 and 55 years. **Results:** Research has shown that the university provided students with various forms of mental health support. During the COVID-19 pandemic, in the 2020/2021 academic year, students and university staff were provided with pedagogical, psychological, and/or psychotherapeutic consultations. Among the main problems reported by students and staff were anxiety, lowered mood, depressive conditions, failure to cope with stress, relationship problems at university, a spectrum of pandemic-related problems, life crises, and discrimination related to sexual orientation. Support was provided via web platforms, social networking websites, and by phone, directly and free of charge. **Conclusions:** The impact of the pandemic has exposed strengths and weaknesses in the management of the mental health support system at the university. It also showed new needs and directions of support. The university has new goals; one of the greatest is preparing students for the challenges of the future.

## 1. Introduction

The SARS-CoV-2 coronavirus emerged in the city of Wuhan, Hubei province, China, in December 2019. On 31 December, China notified the World Health Organization (WHO) of the problem, and on 7 January 2020, the virus was identified. The new disease, which is caused by the coronavirus, has been named COVID-19. The WHO officially declared the COVID-19 outbreak as a public health emergency of international concern on 30 January 2020 [1] and on 11 March 2020 as a global pandemic [2]. The first case of a SARS-CoV-2 coronavirus infection in Poland was reported on 4 March 2020. From 14 March to 20 March 2020, a state of epidemic emergency was in force in Poland, and from 20 March 2020 to 15 May 2022, in accordance with the regulation of the Minister of Health, the state of epidemic was introduced.

The pandemic is still ongoing, and it is manifesting itself in new and more contagious variants, leading to increased infection rates across the globe. More than 660 million people worldwide have already contracted COVID-19, of whom more than 6 million have died [3]. The COVID-19 pandemic proved to be a serious public health threat. With the increasing number of infections with this pathogen, the high death rates, and the lack of capacity of health systems around the world, people feared getting sick, dying, having complications, and coming into contact with others who might be infected.

The effects of a prolonged pandemic have been comprehensively researched and described. By April 2021, there were more than 120,000 publications on COVID-19, of which more than 5000 were on the impact of the pandemic on mental health and how stress resilience was shaped during the COVID-19 pandemic [4].

### 1.1. Student Mental Health before and during the COVID-19 Pandemic—Support Context

The issue of students’ mental health is of great importance to researchers because of the increasing prevalence of mental health issues in the university community. The World Health Organization (WHO) defined mental health as “a state of well-being in which the individual realizes his or her own abilities, can cope with the normal stresses of life, can work productively and fruitfully, and is able to make a contribution to his or her community”. Poor mental health can be linked to fast social change, social exclusion, unhealthy lifestyle, and physical illness [5].

The university years are challenging for many young people. It is a time of transition from late adolescence to adulthood. Importantly, this transition takes place during an extremely vulnerable stage in the human life cycle, when emotional problems and mental disorders often emerge. According to Ronald C. Kessler [6], approximately 75 percent of all mental disorders over a lifetime have their onset before the age of 24. In addition, college years are associated with a significant increase in risky health behaviour, such as excessive use of alcohol and drugs. The college years are not only a time of increased learning and responsibilities, but also adaptation to new surroundings and people, first work experiences, and separation from family and friends. For some, the new challenges motivate them to do more, for others they may cause problems, including those related to mental health. The mental well-being of university students was of concern long before the COVID-19 pandemic [7]. It was suggested that students’ mental health is in “crisis”. The proportion of students who disclose mental health problems to relevant authorities at their university has risen sharply in recent years. Research conducted by the Healthy Minds Network [8] between 2015 and 2021 showed this disquieting trend (Figure 1).

The COVID-19 pandemic, announced in early 2020, has intensified the already difficult mental health situation. To prevent the spread of the virus, numerous restrictions were introduced worldwide. Travel was restricted, quarantines and curfews were imposed, and many sporting, religious, and cultural events were postponed or cancelled. Some countries closed their borders or introduced restrictions on border traffic, including passenger arrivals, and restrictions on border crossers. Passenger temperature controls were introduced at airports and train stations. The pandemic also caused global economic disruption [9] and the greatest disruption to education systems in human history, affecting nearly 1.6 billion students in more than 200 countries. Closures of schools, institutions, and other educational facilities affected more than 94% of the global student population [10]. All of this, combined with prolonged lockdowns and associated social isolation, had a negative impact on the emotions and mental state of entire populations. A worldwide published study on the impact of the COVID-19 pandemic on mental health revealed that the pandemic was associated with increased rates of depression, anxiety, stress, and sleep disorders among different population groups [11,12,13,14,15,16,17,18,19,20]. Marshall et al. [21] estimated that, given the pre-pandemic trajectories, mental health deteriorated significantly: by 8.1%, on average. Young adults and women, i.e., groups who had experienced mental health disorders before the pandemic, were most severely affected.

### 1.2. Mental Health Support at HEIs

Higher education institutions have long provided counselling and mental health support for their students. The majority of HEIs have a mental health policy that outlines how they will approach providing students with mental health services and provisions. Mental health policy documents generally show a commitment to providing a supportive environment for students with mental health difficulties, however this does not always extend as far as providing dedicated on-site mental health services. Universities, as autonomous institutions, differ in how mental health services are organised and provided. Therefore, there are many different forms of wellbeing and mental health support available at universities. Some of these are widely used in many academic centres around the world while others are proprietary solutions derived from the experience and needs of a particular institution.

In most HEIs, support for students with mental health issues is delivered through the Student Services department. Support may be provided through a specific Wellbeing Service, Counselling Service, or Disability Service. HEIs may offer resources such as counselling, workshops on mental health, support networks, and student advising services. According to Goozée et al. [22] web-based alternatives show potential as a novel medium through which students could access support. Online interventions appear to be effective and may therefore provide an alternative to be used by the universities’ student support services (SSS), especially to support students in need of mental health and wellness support, who are unlikely to seek formal help [23,24,25]. Many student unions at HEIs also offer student-led services. Although the students involved are not qualified counsellors, students prefer being consulted by senior students in the times of challenge [26]. According to the profile of the HEI, students may also be provided with the support of a pastoral care team, which may include a personal tutor, a wellbeing advisor, chaplains, and an accommodation team. Additional forms of support may include drugs/alcohol advice, sexual health, access to online wellbeing/self-care support, recommended apps, library wellbeing resources (including online provision), advice from the student finance team, free-of-charge access to sports and exercise facilities, wellbeing programmes, and finally healthcare—nursing and GP support offered on campus or external. To find out what specialised support is offered, students should visit the website of their university.

As an example of good practice, some HEIs developed the role of the Mental Health Advisor. Specialists in this field evaluate how a student’s mental health issues may impact their academic performance. They discuss these with the student and offer methods and interventions to help them advance their academics successfully. MHAs frequently facilitate practical arrangements inside the educational context, enabling them to use context-specific networks to lower educational barriers. Similar to this, MHAs are in a position to guarantee that therapeutic/medical support and practical support are complementary by actively engaging and collaborating with healthcare practitioners [27].

Another form of support offered by HEIs are Mental Health Mentors (MHMs). They only work with students who have received DSA-funded support. Some MHMs work for HEIs, while others are independent contractors or use independent non-medical support worker services. They must be able to comprehend each student’s unique mental health diagnosis, lived experience, and how having a mental illness affects their ability to learn and achieve in their university or university programme. Many students who receive specialised mental health mentoring have been diagnosed with schizophrenia, bipolar illness, eating disorders, personality disorders, sadness, or anxiety. Specialists in mental health mentoring help students understand to the psychological processes at work in their challenges and learn to keep an eye on and control their mental and emotional health [27].

The report from the Royal College of Psychiatrists (RCP), Mental health of students in higher education [28], highlights the concerns related to the provision of support provided to their students by HEIs. The RCP suggests that the student population is diversifying, and part of this variety is putting additional strain on counselling and mental health services, which is driving up demand for university-based counselling and Mental Health Advisor services. Furthermore, universities and other HEIs have undergone changes that have rendered them less equipped to deal with mental illnesses among students. For instance, staff-to-student ratios have fallen as a result of failing to expand staff numbers proportionately to the growth in student enrolment. Academic faculties may have less time for pastoral care because they are constantly under pressure to maintain and enhance their research output as well as to improve their teaching. Finally, resources available for university student support services (SSS) are decreasing, reflecting the general global economic trend. Similar problems are stressed by HUMEN’s research [29]. The results reveal that many UK universities are allocating measly mental health budgets. Data compiled from research and student surveys show a clear disparity between what is now offered and trained for university mental health services against what is needed by students. About half (47%) of students think that having mental health issues while pursuing a higher education had a negative impact on their university experience. Within this difficult context, one recommendation from the RCP suggests self-help programmes, such as web-based interactive cognitive behavioural therapy (CBT), should be used to support students. While in-person counselling should be given priority for those who need it the most, individuals who have less severe requirements may be helped through online self-help, which would ease the load on SSS [22].

Due to the significant increase in the number of young people experiencing mental health issues, ensuring their access to professional help is recognized as an important challenge. Taking up this challenge, Maria Curie-Skłodowska University in Lublin (Poland) has introduced a number of different forms of support. In this paper, we will present a case study on mental health support offered for students and university staff during the COVID-19 pandemic by UMCS. These findings contribute to the literature on mental health support in higher education during the difficult time of pandemic.

## 2. Detailed Case Description

This research covers a descriptive, single-case study. Our case encompasses institutional forms of mental health support at UMCS during the COVID-19 pandemic. To collect data, we used a case study protocol consisting of a set of questions [30]. The main research questions include: (1) What kinds of institutional forms of mental health support were available at UMCS during the COVID-19 pandemic? (2) Did the pandemic affect the mental health support provided at UMCS? Additionally, if yes, then how? In this case study, we used the document research method (we analysed institutional documents, legal acts, statistical summaries, articles, and internet sources) and interviews with UMCS employees. The study was conducted between October and December 2022. Purposive sampling was used in the study. Participants were recruited via convenience sampling methods (personal contacts, phone calls, emails, etc.). Of a total of 32 recruited, 19 participants (12 psychologists, 3 pedagogues, and 4 executives in the organizational units of the UMCS). were included in this study. The reason for the exclusion of the 13 initially recruited university employees was their refusal to participate in the study. Of the respondents, 16 identified as female, 3 identified as male. Ages ranged between 26 and55 years. All participants provided informed consent.

### 2.1. Case Study as a Research Strategy in the Social Sciences

Case study research has a long tradition. It rose in popularity in the social sciences in the early 20th century through the Chicago School. From the mid-1930s to the mid-1960s, as interest in qualitative methods declined, and so did the popularity of the case study. However, the rapid development of qualitative research methodologies, observed since the late 1960s, resulted in a renewed interest in this strategy [30]. The main aim of the case study is to analyse processes with their context in mind since no social phenomenon can be fully understood if it is analysed in isolation from the environment in which it occurs [31]. “The context is deliberately part of the design. As such, there will always be too many ‘variables’ for the number of observations made and so the application of standard experimental or survey designs and criteria is not appropriate. Issues of reliability, validity, and generalizability are addressed but with a different logic and evidence” [32] (pp. 323–324). The aim of the case study may be to explore, describe, and explain [31]. Case studies can be useful in capturing emergent and changing properties of life in organizations [32]. A “case” is generally a bounded entity: a person, organization, behavioural condition, event, or other social phenomena. A case study will consist of a single case or multiple cases and is accordingly labelled a single- or a multiple-case study [31]. Single- and multiple-case studies differ only in the way the research is designed. A case study usually makes use of a variety of research methods and techniques used in both qualitative and quantitative research, such as documentary research, interviews, direct and participant observation, or survey research. It is important to note that in a case study, none of these are prioritized over the others. The use of different research methods and techniques can be considered one of the characteristics of a case study [30].

### 2.2. Maria Curie-Skłodowska University (UMCS) in Lublin, Poland

Maria Curie-Skłodowska University in Lublin is the largest public university in eastern Poland. By May 2022, nearly 260,000 graduates had completed higher education at UMCS. The university provides education at 12 faculties in Lublin and the Puławy Branch, in approximately 80 fields of study and nearly 300 specializations. Every year, it offers candidates new paths of study and improves the already available offer by means of unique and practical specializations. There are currently more than 130 scientific societies and university-wide organizations at UMCS, which bring together active young people who organize interesting initiatives. A total of approximately 17,000 students study at the university, including more than 1560 foreigners from nearly 45 countries. There are also more than 420 doctoral students at the university. The number of students with disabilities is 296, while the number of doctoral students with disabilities is 12. In addition, there are four foreign students with disabilities studying at the university. The university employs 2769 people, including nearly 40 people with disabilities [33].

### 2.3. Forms of Support

The University of Maria Curie-Skłodowska in Lublin offers students various forms of psychological support in learning; personal, social, academic, and professional development; and entering the labour market. These forms are constantly reviewed, updated, expanded, and improved. Information on available forms of support is communicated to students via the USOS website (University Study Support System), educational and administrative platforms, social media, and by academic staff in direct contact. The university uses several channels to gather feedback and ensure the quality of student support: annual surveys, tutors for each year of study, staff consultations for students (also remote), and direct contact with employees. All of these are valuable sources of information about students’ needs and problems. Activities in these channels are systematic, continuous, and comprehensive. In the 2020 pandemic year, a survey was conducted with 1136 students at UMCS. The study focused on students’ needs for psychological support. The obtained data showed 31% of the surveyed students declared a need to talk to a psychologist, 26.1% expressed a desire to participate in skills training, 23.9% wanted to participate in individual psychotherapy, 5.3% preferred group psychotherapy, and 19.7% of the respondents were willing to listen to lectures on coping strategies. Regarding areas of interest, the most frequently mentioned by respondents were stress and coping with stress, depression, mental disorders, occupation, career, work, self-esteem, coping with anxiety, relationships with others, personal development, relationship building, and coping with everyday problems [34]. The next part of the paper will present the forms of mental health support available at UMCS during COVID-19, including the consequences of the pandemic on the scope and forms of provided support.

#### 2.3.1. The Office for Persons with Disabilities and Psychological Support

The Office for Persons with Disabilities and Psychological Support of the Centre for Education and Student Services at Maria Curie-Skłodowska University in Lublin has been operating since 2011. The Office works with students and doctoral students with disabilities and people experiencing mental health crises. It provides counselling and training on effective learning and the use of study time. The unit’s tasks include integrating people with disabilities into the academic environment, equalizing educational opportunities, eliminating barriers, counteracting exclusion, and adapting the didactic process to the needs of students with different health conditions. The main supported groups include students with a recognized disability, chronically ill people, and students and staff experiencing mental health crises. Psychologists and psychotherapists work with the Office to provide study-related counselling and help with various individual problems during on-call hours. All students are assured of discretion and a pleasant atmosphere.

Until the outbreak of the COVID-19 pandemic, two psychologists worked with the Office on a permanent basis. Due to the increased demand for psychological support, an additional three psychologists have been recruited for 2020–2022. Statistics kept by the Office indicate that 75 people were provided with psychological assistance in 2019, 200 people in 2020, 190 people in 2021, and 86 people by February 2022 (data obtained during individual interviews). An almost threefold increase in the need for assistance in 2020 was observed, and as the data show, the trend continues despite the end of the pandemic.

In response to the increase in need, as of 1 October 2022, the Office introduced detailed regulations for the use of free psychological and psychotherapeutic consultations by professionals to overcome mental health crises. This happened during the course of the presented study. At the time of writing this paper, psychological consultations can be used by students, doctoral students, research and teaching staff, and administrative staff of UMCS in mental health crises. Students and doctoral students at UMCS can benefit from support after completing an application form via the MobiSupport app, on the website, face-to-face with a specialist, or at the Office. In the event of an emergency, this formality may be waived or completed at a later date.

A student or doctoral student may benefit from psychological and psychotherapeutic consultations with only one specialist. In the case of complex problems, it is possible to involve an additional expert by arranging it with the Office. The number and frequency of consultations are set after a diagnostic consultation at the first meeting. Consultations take place individually or in groups. The decision to join a group consultation is made jointly with the person receiving support at the individual meeting. Meetings take place in spaces designated by the specialist or at the Office, and in justified cases by telephone or online. Support covers counselling and consultations, including (1) support in coping with current difficulties and problems of psychological nature; (2) support and help with any difficulties in everyday functioning, coping with stress, lack of motivation, relational difficulties, the occurrence of behavioural symptoms, as well as those related to the mental state such as feelings of fear, danger, anxiety, irritability, cognitive disorders, and others; (3) support in further possibilities for help or recommendations of treatment or psychotherapy outside the university. Cooperation between a psychologist or psychotherapist and a client is preceded by a diagnostic consultation. During these 1–2 meetings several issues are defined: the situation of the person in need of help, the aim of the work, the intensity and probable duration of the cooperation, and the applicable rules. Consultations take place regularly, usually once a week, with a maximum of 2–3 times a month. The person receiving psychological support is entitled to five consultations per academic year, and in exceptional situations, the specialist may decide to increase this limit. Increasing the frequency of consultations happens in an agreement between the specialist and the Office. The duration of the session is 50 min. It is the duty of the specialist (psychologist, psychotherapist) to maintain professional secrecy and complete discretion regarding the course of the therapy and the client’s personal data. The specialist is only exempted from professional confidentiality if the student/doctoral student/employee or someone close to them is in danger of losing their health or life.

#### 2.3.2. Sensum—A Support and Psychoeducation Point for Students

A Support and Psychoeducation Point for Students, called Sensum, has operated at the Faculty of Pedagogy and Psychology of Maria Curie-Skłodowska University since 2015. It offers free psychological and pedagogical support to students, doctoral students, and employees of the university. Counselling is provided by six specialists: psychologists, pedagogues, and psychotherapists who are also part of the research and teaching staff of the Faculty of Pedagogy and Psychology. Currently, the Sensum team operates face-to-face and online. During the pandemic, meetings with people in need of support were held via communication platforms and by telephone. Specialists at Sensum provide psychological consultations to people who (1) are facing important life decisions, such as starting a job, formalizing a relationship, starting an independent life, etc.; (2) are looking for a direction in their professional development, or seeking a balance between their professional and non-professional life; (3) want to define their priorities and aspirations in life; (4) want to change something in their life to make it more satisfying and happy; (5) are experiencing difficulties in their daily life, such as stress related to studies and/or work, problems with motivation to act, and difficulties in setting and achieving goals.

Pedagogical, psychological, and psychotherapeutic consultations and individual psychotherapy are carried out with (1) people experiencing stress, anxiety, and other difficult emotions they cannot cope with; (2) people with psychosomatic disorders, affective disorders, obsessive compulsive disorders, social phobia, sleep disorders, and eating disorders; (3) people caught up in difficult relationships;(4) people who want to deal with a painful past; (5) people who experience difficulties in interpersonal relationships; (6) people who need support in personal development, learning about themselves, their needs, and goals, and building adequate self-esteem assertiveness; (7) people who need support and knowledge to fulfil their educational tasks; (8) people who experience difficulties in defining their own identity and making life decisions; and (9) people who need support in the process of expanding their knowledge of parenting or knowledge of the phenomenon of violence [35]. During the COVID-19 pandemic, in the 2020/2021 academic year, the Support Point provided assistance to 132 people (students and staff), which, according to the professionals working there, was a dramatic increase compared to previous years. The assistance consisted of 371 h of individual consultations. Among the main problems reported by students and staff were anxiety, lowered mood, depressive conditions, failure to cope with stress, relationship problems at university, a spectrum of pandemic-related problems, life crises, and discrimination related to sexual orientation.

In the 2021/2022 academic year, support for students, staff, and doctoral students continued to include psychological, educational, and psychotherapeutic counselling. There was also a demand for coaching, psychoeducation, and prevention of risk behaviour and violence. The Sensum team responded by providing lectures and educational meetings on these topics. Students and staff in need of support contacted the specialists directly to arrange assistance according to individual needs. Contact details (email addresses and telephone numbers) were provided on the Sensum website.

#### 2.3.3. Free Assistance via Facebook, Skype, and Telephone, plus Educational Materials

In view of the pandemic, in March 2020, the Institute of Psychology of UMCS offered free psychological support from professionals employed at the Institute. The support was aimed at people who were experiencing anxiety and insecurity due to the COVID-19 pandemic outbreak and had lowered mood and feelings of isolation and alienation. Free support was provided by eight psychologists via Facebook and Skype messaging, as well as by telephone. Contacts were posted on each faculty’s website and students from various faculties were seeking help. Among the most frequently reported problems were difficulties adjusting to the demands of remote learning during the pandemic, such as problems with public speaking; emotional difficulties (anxiety, lowered mood) related to the pandemic; problems in the social sphere, conflicts and tensions in interpersonal relationships during the pandemic; problems resulting from the loss of a job during the pandemic and the need to choose a new career path; and family difficulties exacerbated during the pandemic. Estimates from psychologists show that between March 2020 and July 2021, between a dozen and up to several hundred people benefited from the support. Individual support ranged from one to dozens of online meetings or phone calls.

The “Friendly University” project, started in the 2014/2015 academic year at UMCS, aims to bring mental health issues to the attention of the academic community and to build a support network for students encountering mental health difficulties. The project website has brochures providing information on the specifics of the various problems, as well as guides on good practices in supporting individuals in difficulty. During the COVID-19 pandemic, the project website featured informational materials showing how to take care of one’s health during the pandemic, how to organize one’s time, and how to support people who are in crisis [36].

#### 2.3.4. Academic Support Centre

The Academic Support Centre (ACW) is a university-wide unit based at the Faculty of Pedagogy and Psychology of UMCS. At the ACW, appropriately trained fourth- and fifth-year psychology students provide assistance to students in crisis. Statutory goals of the organization include (1) providing psychological support to students experiencing psychological difficulties, (2) providing information about specialists and facilities offering free psychological assistance, (3) promoting mental health among students, (4) organizing educational and scientific events, (5) conducting research projects, (6) and promoting psychological knowledge [36]. ACW members are on call to provide consultations. Students can sign up for consultations through social media and by email. The organization guarantees anonymity, patience, understanding, and trust to those seeking help. During the COVID-19 pandemic, the ACW did not hold consultations and limited its activities to social media campaigns. Currently, consultations take place face-to-face as before the pandemic.

#### 2.3.5. Solutions Introduced after the Pandemic

In response to the demand for knowledge on the mental functioning of an individual in a difficult, crisis situation, UMCS co-financed the publication of a series of e-books appearing successively from 2021 under the common title *Coping Strategies During a Pandemic and Beyond*. The educational materials are available free of charge on the website of the ArchaeGraph Publishing House [37]. The issues presented in the publications are intended for people of all age groups: children, adolescents, and adults; there is also content for parents.

In 2021, the Optimum Team for optimizing the educational conditions for students with individual educational needs was established at UMCS. Its activities are aimed at ensuring optimal conditions (including psychological conditions) for persons with disabilities to participate fully in the university enrolment process, then in the educational process, as well as in academic activities. The Team supports the implementation of the University’s accessibility strategy in its broadest sense. It is made up of four specialists, academic staff employed by the Faculty of Pedagogy and Psychology. They act as advisors for optimizing learning conditions for students with visual, auditory, and mental impairments, and students onthe autism spectrum. Optimum deals, among other things, with educational support for UMCS staff on solutions to help students with mental health problems (depression, mood disorders, mental illness, etc.) function academically. Specialists share their knowledge on how to contact students, enforce requirements, or communicate in class.

The sensitization to mental health problems that occurred during the COVID-19 pandemic (Table 1) resulted in a number of socio-cultural initiatives and artistic events (film screenings and photography exhibitions) at UMCS addressing the issue of human functioning in conditions of isolation, uncertainty, and loss. “Psychoperformatics—books that change lives” was a series of meetings held since 2021 that made use of the therapeutic function of art. Selected literary works were read out in a theatrical form, which inspired the participants to reflect on current but also timeless problems relating to the human condition, the relationship with the world of objects and people, or significant events—both on a micro and macro scale.

In 2022, work was completed on two strategic documents on increasing the University’s accessibility for people with disabilities. The first is the “Social Responsibility Strategy for Persons with Disabilities at the University of Maria Curie-Skłodowska in Lublin. Consciously and Responsibly—Education in Diversity”, the second is the “Strategy for Universal Design at the Maria Curie-Skłodowska University in Lublin”. Both documents take into account the lessons learned from the COVID-19 pandemic, emphasizing the need to support students’ mental health by building a welcoming and safe educational environment.

## 3. Discussion

### 3.1. Principal Results and Comparison with Prior Findings

The incidence of mental illness, mental distress, and low well-being among students in higher education worldwide is increasing and is high relative to other social groups. Therefore, supporting students’ mental health requires an effective response across the higher education sector and beyond, so that students receive the support they need at difficult times. The examples of good practice presented in this article, which were implemented at UMCS during the COVID-19 pandemic and have been continued to the present day, address these needs. Student support is comprehensive and systematic and is provided both by various university-wide units and by the students themselves.

Students and university staff can benefit from the Psychological Counseling Service set up within the Disability and Psychological Support Office of the Centre for Learning and Student Services. It supports all students at UMCS who are in a personal crisis. The counselling centre offers the services of psychologists, educational counsellors, and therapists. Psychological support is provided by the University’s employees at Sensum—the Support and Psychoeducation Point, operating at the Faculty of Education and Psychology at UMCS as well. Students can also count on support offered within their faculties from, among others, year tutors, vice-deans for student affairs, and from administrative staff dealing with student services. Additionally, there is support provided by the staff of the Competence Development Office, the Office of Student Affairs, the Office for Studies and International Students, or the activities of the Rector’s Plenipotentiary for Student Affairs and the Vice-Chancellor for Students and Educational Quality. A unique proposal addressed to students is the activities of the Academic Support Centre, whose members as final-year students of Psychology are trained and prepared to provide peer support. They are regularly on call for help, run educational and preventive campaigns, and actively participate in training and workshops, constantly improving their qualifications.

For healthcare reasons, most of the research conducted in groups of students focused on identifying risk factors [38,39,40] and the effects of the COVID-19 pandemic on young people’s mental health [41,42]. However, some researchers also focused on describing forms of support for students and HEI staff in mental health crises. Various forms of student support, mainly information campaigns in the form of provided assistance, psychological consultations, and support groups, were also carried out by other universities in Poland [43,44] and worldwide [12,45,46].

A study initiated by the Patient Ombudsman [43] in 40 higher education institutions in Poland showed that university authorities notice mental health problems among both students and staff. Polish HEIs strive to provide comprehensive psychological support, often enriched by cooperation with medical institutions, NGOs, and associations that provide specialist psychological support. Among the HEIs that responded to the Patient Ombudsman, fifteen have a mental health counselling centre within their structure and one is planning to establish a counselling centre. Preventive workshops, training, and lectures are organized. Many institutions (just as UMCS in Lublin) involve their staff and students in supporting students who need it. Some universities use fourth- and fifth-year psychology students to provide psychological support. Collegiate support groups are organized for students who are reluctant to come forward for professional help. Students who have experienced a mental health crisis share their experiences and help them to find specialist help. The universities provide coaching, psychotherapy, environmental interventions, and anti-stress therapies. Support groups for people with disabilities are created. Activities related to psychoeducation and the promotion of a healthy lifestyle, pro-health education, and prevention activities are extremely important. At some universities, sports activities are organized as a way of relieving negative emotions.

In most HEIs in England, support for students with mental health issues is delivered through the Student Services department. Support may also be provided through a specific Wellbeing Service, Counselling, or Disability Service. HEIs may provide counselling, student advice service, support networks, mental health workshops, or other resources. Students’ unions may also provide student-led services such as peer support groups and advice lines [12] (p.13).

In September 2020, The American Council on Education (ACE) surveyed 268 university presidents to find out how the institutions they led were responding to the challenges of COVID-19 and to better understand both the immediate and long-term impact of the pandemic on higher education. The results obtained by ACE showed that the issue of student mental health during the COVID-19 pandemic was exigent for 70% of US university presidents. In order to respond to the growing mental health concerns of their students, university leaders implemented a variety of strategies to support the mental health and well-being of their academic community. The most common forms of support appeared to be virtual or teletherapy services and telepsychiatry (59%). Fewer than 50% of the presidents revealed that their university introduced “new student engagement strategies to provide students with resources on mental health and well-being” (47%) and “expanded campus access to digital mental health programs and promotion platforms” (43%). Among other support strategies, presidents reported that they examined the institution’s policies, programs, and systems that support student mental health (29%); increased the number of counselling centre staff (29%); targeted the outreach of mental health resources to students disproportionately impacted by COVID (21%); presented COVID-19-specific training to faculty and staff about supporting student mental health during COVID (21%); conducted a campus-wide assessment of student mental health needs (18%); and appointed a task force or committee focused on identifying mental health needs and strategies (12%). Unfortunately, 9% of the surveyed presidents stated that their university had not implemented any support strategies during this difficult pandemic time [45].

In 2021, one year after the outbreak of the pandemic, the International Association of Universities (IAU) conducted the second global survey of the impact of COVID-19 on higher education. The total number of HEIs that completed the whole survey was 469, in 112 countries and territories. The results show that the percentages of HEIs offering physical and mental support for students are similar for private and public HEIs (79% vs. 77% for physical health and 82% vs. 80% for mental health). However, more private HEIs reported student satisfaction (41% fully and 54% somewhat vs. 35% fully and 50% somewhat for physical health and 42% fully and 50% somewhat vs. 31% fully and 56% somewhat for mental health). Half of the participating institutions increased their level of support for physical and mental health and only a few reported a decrease in these services, meaning that for the rest, service levels remained the same as before the pandemic. The authors of the study emphasized that at this stage of the study, they could not indicate what type of services were offered, nor whether the increased health services were temporary due to the pandemic or if they would remain in the offer of a particular university for a longer period of time. According to the researchers, this could also be linked to the different health systems in different countries. Perhaps in countries where healthcare systems are readily available to all, the role of the university is less important, whereas in countries where access to healthcare is limited, the responsibility for ensuring the well-being of staff and students is the responsibility of the universities [46].

### 3.2. Limitations

Described findings contribute to the literature on mental health support in higher education during the difficult time of pandemic. However, several important limitations of the study can be stressed. First of all, in this research efforts were made to reach every person responsible for providing support to students at Maria Curie-Skłodowska University (UMCS) in Lublin, Poland. In some cases, individuals refused to participate in the research. Furthermore, we faced difficulty in accessing complete data on offered support. Psychological support at UMCS provided by professionals (psychologists, psychotherapists, and educators) is voluntary, frequently extemporary, and unpaid (except for the Office for Persons with Disabilities and Psychological Support, where specialists are paid). Employees have professional autonomy in the area of support provided; they are mostly not required to keep detailed documentation or report it to their superiors. Specialists are also bound by the duty of professional confidentiality, which has significantly limited the scope and detail of the information obtained.

## 4. Conclusions

Over the last three years, the world has been impacted in many ways by the pandemic. Besides the health problems and the strain on the health systems around the world, the pandemic has affected the economic, social, and emotional well-being of society. The socio-cultural and economic system we knew has been destabilized and the consequences may be long-term. Students at universities around the world have also been affected by COVID-19. Remote learning and social distancing measures implemented at educational institutions have radically changed campus life. In addition to the changes experienced by the general public, students have also experienced those that have negatively affected the methods, forms, and outcomes of their studies. In addition, the realization that the pandemic may affect young people’s future educational opportunities, employment prospects, and financial stability causes ongoing tensions, and prolonged stress, and intensifies students’ already difficult emotional and psychological situation.

Several key recommendations emerge from the research. In order to control the scope and quality of the provided support, as well as to effectively monitor students’ needs, a mandatory procedure should be introduced for professionals to report their work. Reporting must include both quantitative and qualitative data, without confidential information. It is also necessary to introduce a system for assessing the quality of the support by the students who benefit from it and to strengthen the coordination of forms of support between the various units operating within the university. So far, individual counselling has mostly covered emotional problems related to stress, anxiety, and difficulties in everyday functioning. At this stage, it seems that the university’s activities should be complemented by broader preventive measures to reduce the occurrence of psychological problems among students in the future. Additional educational activities (such as information campaigns, workshops, and lectures) are necessary to teach about media (the impact of digital media on health, how to balance “online life” and “offline life”), help develop the skills to cope with difficult and stressful situations, to adapt to change, and to build satisfactory interpersonal relationships.

It is also essential to strengthen peer support and build an authentic university community through participation in social, scientific, cultural, and sporting events. The goal of the university in times of uncertainty must be to mentally prepare students for the challenges ahead.

The post-pandemic period seems to be a critical time to intervene in cases of university students’ mental health crises. It is our hope that the findings in the present paper will inform universities, healthcare professionals, and government agencies about the principal issues faced by students and how they can find effective solutions. The views presented are part of a widespread and ongoing reflection on the goals of today’s universities and how they are achieving them. Faced with the need to cope with challenging realities and widespread change, academia should constantly self-reflect.

## Figures and Tables

**Figure 1 ijerph-20-04969-f001:**
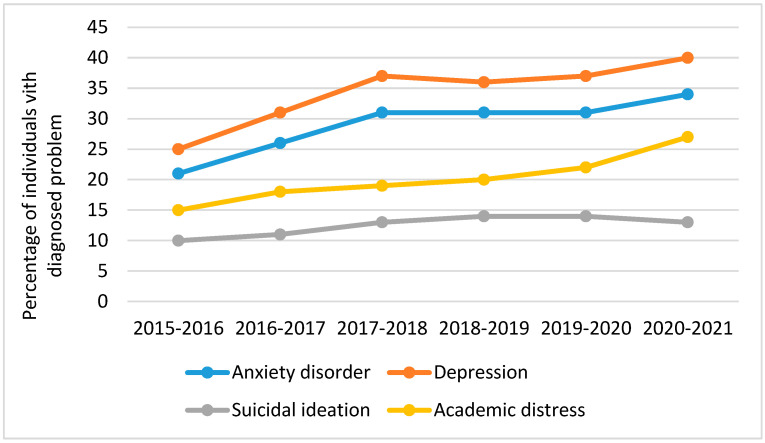
Healthy Minds Study 2015–2021.Source: based on the data retrieved from https://www.datawrapper.de/_/oQcy0/ (accessed on 3 January 2023).

**Table 1 ijerph-20-04969-t001:** The forms of mental health support available at UMCS during and after COVID-19 pandemic.

Provisions/Services Offered to Students & Staff		Recipient of the Support	Type of Intervention	Form of Intervention
1.	The Office for Persons with Disabilities and Psychological Support	students with a recognized disability, chronically ill, and students and staff experiencing mental health crises	- counselling and training on effective learning and the use of study time; - integrating people with disabilities into the academic environment, equalizing educational opportunities, eliminating barriers, counteracting exclusion; - adapting the didactic process to the needs of students with different health conditions.	- individual; - group; - face-to-face; - online; - free of charge.
2.	Sensum—a Support and Psychoeducation Point for Students	- students, staff, and doctoral students: - experiencing stress, anxiety; - with psychosomatic disorders, affective disorders, obsessive compulsive disorders, social phobia, sleep disorders, and eating disorders; - caught up in difficult relationships; - who want to deal with a painful past; - who experience difficulties in interpersonal relationships; - who need support in personal development; - who need support in their educational tasks; - who experience difficulties in defining their own identity and making life decisions; - who need support in the process of expanding their knowledge of parenting or knowledge of the phenomenon of violence.	- pedagogical, psychological, psychotherapeutic consultations - individual psychotherapy	- individual; - group; - face-to-face; - online; - free of charge.
3.	Free assistance via Facebook, Skype, and telephone, plus educational materials	students, staff, and doctoral students who were experiencing anxiety and insecurity due to the COVID-19 pandemic outbreak, and had lowered mood, and feelings of isolation and alienation	pedagogical, psychological, psychotherapeutic consultations	individual;online via Moodle/MsTeams platform;via Facebook a Skype messaging, and by telephone;free of charge.
4.	Academic Support Centre	students in crisis	student-to-student intervention aimed at: - psychological support to students experiencing psychological difficulties—providing information about specialists and facilities offering free psychological assistance; - promoting mental health among students.	- individual; - by telephone; - face-to- face; - free of charge.
5.	A series of e-books: Coping strategies during a pandemic and beyond	the entire university community	- a series of publications answering the most common and topical questions addressed to the psychologist-practitioner, psychotherapist during the pandemic, - provide self-development, educational and guidance material.	- available on the website of the ArchaeGraph Publishing House; - free of charge.
6.	The Optimum Team	students with a recognized disability, chronically ill, and students experiencing mental health crises	ensuring optimal conditions (including psychological conditions) to participate fully in the university enrolment process, then in the educational process, as well as in academic activities	- individual; - face-to-face; - online; - free of charge.

## Data Availability

Data supporting the reported results can be obtained by email from the corresponding author.

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
