# Peer review of "Mental Health Support in Higher Education during the COVID-19 Pandemic: A Case Study and Recommendations for Practice"

_ijerph, 2023, doi:10.3390/ijerph20064969_

Round 1
Reviewer 1 Report
Title: The impact of the COVID-19 pandemic on mental health support in higher education: a case study and recommendations for practice
Article Type: Article
Summary
In this case study, the authors examine the impact of COVID-19 pandemic on mental health in higher education in Poland. The results of this case study indicated that the university provided students with various forms of mental health support via web platforms, social networking websites, and even by phone, directly and free of charge. Moreover, the impact of the pandemic has exposed strengths and weaknesses in the management of the mental health support system at the university.
Evaluation
I acknowledge the authors' effort in conducting the study. However, the topic has already been well described in the literature and even in Poland, so couldn’t add any new findings to the literature. So, I think this manuscript hasn’t appropriate acceptability for publishing in the Journal. In this area there are several research works that all have investigated the current research topic. For example,
1- Wieczorek T, KoÅ‚odziejczyk A, CiuÅ‚kowicz M, Maciaszek J, Misiak B, Rymaszewska J, SzczeÅ›niak D. Class of 2020 in Poland: Students’ mental health during the COVID-19 outbreak in an academic setting. International journal of environmental research and public health. 2021 Mar 11;18(6):2884.
2- Rogowska AM, Ochnik D, Kuśnierz C, Chilicka K, Jakubiak M, Paradowska M, Głazowska L, Bojarski D, Fijołek J, Podolak M, Tomasiewicz M. Changes in mental health during three waves of the COVID-19 pandemic: A repeated cross-sectional study among Polish university students. BMC psychiatry. 2021 Dec 15;21(1):627.
However, the below points and suggestions should be addressed by the authors, in order to improve the quality of the manuscript for submitting it to other Journals.
Please add some information about your case in the abstract, for example, write a little about the participants, their age, their gender, etc.
Please write the results of the study in detail in the abstract.
Please write the exact date and period for the study in the abstract.
In Figure 1, please add error bars and also the label of Y axis, I mean the Figure is not complete and so need to be revised.
You wrote partially a good introduction, but the research gap and also the research question are still vague.
2.1. Case study as a research strategy in the social sciences. I think this is not need to write and define the case studies here in the current study.
I think, it is needed to describe the results of the study in a better form, it is vague at the present form.
Discussion is good but it is better to add the studies limitation too.
Author Response
Response to Reviewer 1 Comments
Point 1: I acknowledge the authors' effort in conducting the study. However, the topic has already been well described in the literature and even in Poland, so couldn’t add any new findings to the literature. So, I think this manuscript hasn’t appropriate acceptability for publishing in the Journal. In this area there are several research works that all have investigated the current research topic. For example,
- Wieczorek T, KoÅ‚odziejczyk A, CiuÅ‚kowicz M, Maciaszek J, Misiak B, Rymaszewska J, SzczeÅ›niak D. Class of 2020 in Poland: Students’ mental health during the COVID-19 outbreak in an academic setting. International journal of environmental research and public health. 2021 Mar 11;18(6):2884.
- Rogowska AM, Ochnik D, Kuśnierz C, Chilicka K, Jakubiak M, Paradowska M, Głazowska L, Bojarski D, Fijołek J, Podolak M, Tomasiewicz M. Changes in mental health during three waves of the COVID-19 pandemic: A repeated cross-sectional study among Polish university students. BMC psychiatry. 2021 Dec 15;21(1):627..
Response 1: After careful consideration of the listed articles, we would like to point that both present studies from defferent perspective than the reviewed one. Wieczorek et. al (2021) describe general aspects of mental health of students and provide data on students’ worsened mental health due to the COVID-19 pandemic. Similarly, research conducted by Rogowska et.al (2021) aims to examine changes in mental health among university students during Covid-19 pandemic. The aim of the research described in submitted article was different. We tend to describes the forms of psychological support offered by Maria Curie-SkÅ‚odowska University in Lublin during that difficult time as an example of good practices and real respond to the need stressed in both listed articles – “recommendations for early psychosocial interventions among students”. The described in our aricle case has a specific character due to the organizational solutions introduced at the university. In our opinion these findings contribute to the literature on mental health support in higher education during the difficult time of pandemic.
Point 2: Please add some information about your case in the abstract, for example, write a little about the participants, their age, their gender, etc. Please write the results of the study in detail in the abstract. Please write the exact date and period for the study in the abstract.
Response 2: The abstract is completed with the suggested information. Thank you for highlighting the missing data.
Point 3. In Figure 1, please add error bars and also the label of Y axis, I mean the Figure is not complete and so need to be revised.
Response 3: Indeed, Figure 1 does not contain error bars. I am unable to include this data in the graph due to the fact that the source from which I drew this data does not contain this information. The author did not include them in his research. I have added the label of Y axis and the detail in the cited source: “Source: based on the data retrieved from https://www.datawrapper.de/_/oQcy0/(accessed on Jauary 3, 2023)”.
Point 4: You wrote partially a good introduction, but the research gap and also the research question are still vague.
Response 4: Paragraph has been added to the ending part of Intorduction (p. 6).
“According to the research review presented, there has been a significant increase in the number of young people experiencing mental health issues for many years now, which has been exacerbated during the difficult times of the COVID-19 pandemic. Supporting young people’s mental health and ensuring their access to professional help is therefore an important challenge. Taking up this challenge, Maria Curie Sklodowska University in Lublin (Poland) has introduced a number of different forms of support. In this paper, we will present a case study on mental health support offered for students and university staff during the COVID-19 pandemic by UMCS. These findings contribute to the literature on mental health support in higher education during the difficult time of pandemic”.
Point 5: 2.1. Case study as a research strategy in the social sciences. I think this is not need to write and define the case studies here in the current study.
Response 5: In our concept of the paper the definition of case study is concise and is intended to illustrate how method is understood in this particular study, i.e., as a method of describing processes in organizations. If it is suggested we are able to remove that part from the article.
Point 6: I think, it is needed to describe the results of the study in a better form, it is vague at the present form.
Response 6: The idea of the paper was to decribe precisely every particular form of support offered by our Univeristy to students and staff, as an example of so called “good pracitces”. That is why , the form of presenting the results of the study in this way is relevant for us. All forms of support are listed and described in separate sections and can serve as examples.
Point 7: Discussion is good but it is better to add the studies limitation too.
Response 7: Studies limitation paragraph has been added - 4.2. Limitations (p. 13)
Thank you so much for your effort and valuable suggestions to improve the paper

Reviewer 2 Report
There is no doubt that the manuscript has some practical significance. However, may I suggest to the authors:
1. The authors should tell readers the motivation and reason why they decide to complete the manuscript in the Introduction section.
2. Limitation should be put in Limitation section rather than “materials and method” section.
3. The author should provide some details about patticipants, such as gender, age.
4. There is number error (line 369) in “3.1 The Office for Persons with Disabilities and Psychological Support”.
5. Most importantly, the emphasis of the manuscript should be the impact of the COVID-19 pandemic on mental health support according to title, but the author prefers to focus on mental health rather than mental health support in the introduction, which is obviously inappropriate, and the expression of the title is also inappropriate, I suggest changing it to “ Mental health support in higher education during the COVID-19 pandemic: a case study and recommendations for practice”.
Author Response
Response to Reviewer 2 Comments
Point 1: The authors should tell readers the motivation and reason why they decide to complete the manuscript in the Introduction section
Response 1: The concept of the Introduction was to give the reader deatailed insight into the general aspects of mental health of students and provide data on students’ worsened mental health due to the COVID-19 pandemic. It was to justify the need of the research. We tend to describes the forms of psychological support offered by Maria Curie-SkÅ‚odowska University in Lublin during that difficult time as an example of good practices and real respond to the need stressed in many papers i.e. supporting young people’s mental health and ensuring their access to professional help. According to the comment the last parragraph of the Intorcuction is rephrased.
Point 2: Limitation should be put in Limitation section rather than “materials and method” section.
Response 2: Study limitations subsection has been added - 4.2. Limitations (p. 13)
Point 3. The author should provide some details about patticipants, such as gender, age.
Response 3: The subsection 2. Materials and Methods is completed with the suggested information. Thank you for highlighting the missing data.
Point 4: There is number error (line 369) in “3.1 The Office for Persons with Disabilities and Psychological Support”.
Response 4: The number is corrected.
Point 5: Most importantly, the emphasis of the manuscript should be the impact of the COVID-19 pandemic on mental health support according to title, but the author prefers to focus on mental health rather than mental health support in the introduction, which is obviously inappropriate, and the expression of the title is also inappropriate, I suggest changing it to “ Mental health support in higher education during the COVID-19 pandemic: a case study and recommendations for practice”.
Response 5: The concept of the Introduction part was explained in Response 1. Of course we totally agree with your suggestion to change the title. Now it it as suggested: Mental health support in higher education during the COVID-19 pandemic: a case study and recommendations for practice.
Thank you so much for your effort and valuable suggestions to improve the paper.

Round 2
Reviewer 1 Report
I appreciate the author's efforts for the improvement of the article's quality, however, I think the article's quality is not still good, so I prefer to reject the manuscript again.
Good luck with your future research!
Author Response
Dear Reviewer,
Thank you again for your effort to assess the manuscript. It has been revised according to the suggestions made in the reviews. In the lack of a substantive response to the proposed version of the manuscript, we are unable to address the negative assessment of the manuscript contained in the comments to the Authors .
Best regards,
Authors
Reviewer 2 Report
As the focus of this article is on mental health support rather than mental health, it is suggested that the authors appropriately add a review of mental health support, and mental health support during the pandemic in the introduction.
Author Response
Response to Reviewer 2 Comments
Point 1: As the focus of this article is on mental health support rather than mental health, it is suggested that the authors appropriately add a review of mental health support, and mental health support during the pandemic in the introduction.
Response 1: According to suggestion we modified the Introduction part. Point 1.1 as outlining the context for the need for support after revision is limited to the most important issues and intitled: “Student mental health before and during the COVID-19 pandemic –support context.” Due to the fact that references to different forms of mental health support during the pandemic are included in the discussion of the results, we have dropped them from the Introduction section to avoid repetition.
Thank you once again for your effort and suggestions to improve our manuscript.